# Effect of 3D Food Printing Processing on Polyphenol System of Loaded *Aronia melanocarpa* and Post-Processing Evaluation of 3D Printing Products

**DOI:** 10.3390/foods12102068

**Published:** 2023-05-20

**Authors:** Quancheng Zhou, Xijun Nan, Shucheng Zhang, Liang Zhang, Jian Chen, Jiayi Li, Honglei Wang, Zheng Ruan

**Affiliations:** 1State Key Laboratory of Food Science and Technology, School of Food Science and Technology, Nanchang University, Nanchang 330047, China; 2Department of Food Science, School of Agricultural Engineering and Food Science, Shandong University of Technology, Zibo 255049, China; a985354232@163.com (X.N.);; 3Yantai Shuangta Food Co., Ltd., Yantai 265404, China; 4Arkhum Health Technology Co., Ltd., Zibo 255035, China; 5Shandong Jiucifang Biotechnology Co., Ltd., Zibo 256102, China

**Keywords:** 3D food printing, *Aronia melanocarpa* polyphenol, gel, printability, applicability, storage

## Abstract

*Aronia melanocarpa* polyphenols (AMP) have good nutritional values and functions. This study aimed to explore the printability and storage properties of AM gels in 3D food printing (3DFP). Therefore, 3DFP was performed on a loaded AMP gel system to determine its textural properties, rheological properties, microstructure, swelling degree and storage performance. The results revealed that the best loading AMP gel system to meet the printability requirements of 3DFP processing was AM fruit pulp:methylcellulose:pea albumin: hyaluronic acid = 100:14:1:1. Compared with other ratios and before 3DFP processing, the best loading AMP gel system processed by 3DFP exhibited the lowest deviation of 4.19%, the highest hardness, the highest elasticity, the least adhesion, a compact structure, uniform porosity, difficulty in collapsing, good support, a high degree of crosslinking, and good water retention. Additionally, they could be stored for 14 d at 4 °C. After post-processing, the AMP gel had a favorable AMP release rate and good sustained release effect in gastrointestinal digestion, which conformed to the Ritger–Peppas equation model. The results revealed that the gel system had good printability and applicability for 3D printing; as well, 3DFP products had good storage properties. These conclusions provide a theoretical basis for the application of 3D printing using fruit pulp as a raw material.

## 1. Introduction

Three-dimensional food printing (3DFP) technology is a type of processing technology that can form 3D structures by stacking edible materials [1]. Recently, 3DFP has been increasingly favored by researchers owing to its personalization and novelty. The composition and processability of 3DFP ink, particularly the rich ink in fruits and vegetables, have been key to the development of the 3DFP technology [2]. However, fruits may be considered among the hardest materials for 3DFP because of their low viscosity [3]. A common method used in research involves converting fruits and vegetables into fruit and vegetable powders. For example, Chen et al. [4,5]. studied the printing characteristics and sensory evaluation properties of five different animal and vegetable proteins as a matrix and fruit powder composite printing ink. Nevertheless, these processing methods cannot retain the original nutrients in the fruit. Furthermore, few studies have been conducted on the application of 3D printing in the direct processing of fruits and vegetables [3]. Eating more nutritious fruit can effectively regulate body health and reduce the risk of diseases [6]. Consequently, research on the printability and applicability of fruits as a raw material is of great significance for theoretical and practical applications in the field of 3D printing. Moreover, the fruit of functional components can also be well preserved and play their functions, the formula can be adjusted according to its nutritional characteristics and physical properties, and the model can be designed and printed to meet the precise nutritional and personalized needs of people. The fruit of *Aronia melanocarpa* (AM) is rich in polyphenols. AM polyphenols (AMP) have antioxidants, and lipid-lowering, anti-inflammatory, and anticancer effects [7]. Additionally, AMP can improve the intestinal flora and assist in lowering blood glucose and pressure [4]. The demand for AM is gradually increasing owing to the good nutritional value and function of AM. However, the common AM processing method is dry, whereas other AM processing methods and applications have been studied less extensively.

Methyl cellulose (MC) is a safe, non-toxic food additive with good biocompatibility and gelling properties; it has been used as a cytoskeleton in disease treatment [8]. MC has good water solubility and film-forming properties, and it can be used as an edible membrane material [9]. Additionally, it can be made into gels [10], which can be used as 3D printing materials, in theory; nonetheless, studies in the field of 3DFP processing are limited. Hyaluronic acid (HA) is a straight-chain polysaccharide widely found in various human and vertebrate tissues. It has good biocompatibility and physiological functions, including water retention, thickening, and delayed aging [11,12]. Some studies have added HA to nutritional preparations to improve their rheological properties [13]; however, HA as a functional factor, added to the 3DFP printing system, has not been reported. Pea albumin (PA) has good gel properties, solubility, and emulsification, as well as certain functional properties, such as improving obesity and mediating metabolism [14]. Nonetheless, studies on PA and its scope of application are limited.

Accordingly, this paper aimed to build an AM/MC gel system suitable for 3D printing processing. By discussing the influence of 3D printing processing on the physicochemical properties, structure, and physical properties of the gel system, the physical property requirements of AM for realizing 3D printing were revealed, and PA and HA were added to the printing system. Hence, the objective of this study was to explore the effects of PA and HA on the physicochemical properties, structure, and physical properties of the gel system.

## 2. Material and Methods

### 2.1. Preparation of AM 3D Printing Inks and Printing

In this experiment, 100 g of Aronia melanocarpa (AM) fruit pulp was mixed with methylcellulose (MC), pea albumin (PA), and hyaluronic acid (HA) in specific proportions (refer to Table 1 for the composition of each ingredient) [13,15]. The fruit pulp was sourced from Fukangyuan no. 1, Zibo Forestry Protection and Development Center, China. The mixture was then prepared as an ink for the extrusion-based 3D food printer (FOODBOT-S, Hangzhou Printing Technology Co., Ltd., Beijing, China).

The 3D printer used the provided “letter D” model to print the samples. The printing parameters included a 1.20 mm inner diameter for the printing needle, a 3 mm printing height, a 30 mm/s moving rate, and a 1.20 mm layer height.

Before and after the printing process, the samples were placed in disposable petri dishes, sealed with plastic wrap, and tested for various indicators within 1 h. Furthermore, the AM gel powders were obtained after lyophilization (−50 °C, vacuum degree is 22) by a lyophilizer (Beijing Boyikang Experimental Instrument Co., LTD., FD-1A-50, Beijing, China) and stored in −20 °C refrigerator for future use.

### 2.2. Study on the Influence of 3D Printing Materials on Formability

The formability of the 3D printed food products was evaluated through visual analysis, focusing on the smoothness of the 3D printing process, the printed shape, and the forming effect of the printed objects. This qualitative assessment aimed to identify any defects or inconsistencies in the printed structures and to determine whether the chosen materials and parameters resulted in successful printing outcomes.

### 2.3. Determination of Printing Deviation of 3DFP Products

A micrometer caliper (VolScan Profiler; Stable Micro Systems, Surrey, UK) was used to determine the maximum width of the 3DFP products. This was compared with the model design width and calculated width deviation. Width deviation was calculated using the following equation [16].
(1)$$\mathrm{Departure}\;(\%)=\frac{3\mathrm{DFP}\;\mathrm{products}\;\mathrm{measurement}\;\mathrm{width}\;-\;\mathrm{Model}\;\mathrm{design}\;\mathrm{width}}{\mathrm{Model}\;\mathrm{design}\;\mathrm{width}}\;\times \;100\%.$$

### 2.4. Texture Analysis

A texture analyzer (TA-XT 2, Stable Micro Systems, Ltd., Surrey, UK) measured the texture of the 3DFP materials and products (hereinafter referred to as samples) at 25 ± 0.2 °C using a disc type probe (P/75); additionally, the TPA mode was used to measure the hardness, springiness, chewiness, and adhesiveness of samples. The probe compressed the samples with the strain set to 70%. The initial distance between the plate and probe was set to 3.0 cm, and the down, compression, and return speeds were set to 5, 1, and 1 mm/s, respectively. The starting point induction force was 5 g, and the interval between the two compression times was 5 s. Each sample was tested in triplicate with different formulations [1].

### 2.5. PCA (Principal Component Analysis)

In order to obtain AM printing products with good physical properties, principal component analysis and a membership comprehensive scoring method were used to determine the comprehensive score of AM printing products [17]. According to the importance of the physical performance indicators of AM printing products, multiple indicators are simplified into a single indicator, and the weight of the importance of each indicator is determined according to the principal component analysis. In the practical application process, the greater the hardness, elasticity, and chewability, the better, with the membership degree calculated according to Formula (2); the smaller the adhesion, the better, with its membership calculated according to Equation (3) [18].
(2)$$P=\frac{A_{i}{\;-\;A}_{\min}}{A_{\max}{\;-\;A}_{\min}}\;\times \;100\%$$
(3)$$P=\frac{A_{\max}{\;-\;A}_{i}}{A_{\max}{\;-\;A}_{\min}}\;\times \;100\%$$
where P is the index membership degree; A_i_ is the index value; A_min_ is the minimum value of the same index; and A_max_ indicates the maximum value of the same indicator. The comprehensive physical properties of the film are divided into S and calculated according to Formula (4).
S = a P_1_ + b P_2_ + c P_3_ + d P_4_(4)

In the formula, P_1_, P_2_, P_3_, and P_4_ are the membership degrees of four indexes, namely deviation, hardness, elasticity, chewability, and adhesion. a, b, c, and d are the weights of the four indicators, respectively. The weights obtained are substituted into the formula for calculation.

### 2.6. Characterization of the Rheological Properties of Samples

The rheological properties of the samples were characterized using a Kinexus Pro rheometer (Malvern Instruments Ltd., Worcestershire, UK) operated with a serrated parallel plate geometry with a diameter of 40 mm and gap of 1 mm. Subsequently, the frequency sweep test was performed at 25 °C in the range of 0.10–10.00 Hz at a strain of 0.20%. Each test was performed in triplicate [19].

### 2.7. Fourier Transform Infrared (FTIR) Spectroscopy

The method of Xiao [20] was slightly modified. Before and after printing, 2 mg of the sample powder was accurately weighed and thoroughly mixed with 100 mg of potassium bromide powder. The powder was evenly ground using a spherical grinding machine. After pressing, the samples were loaded into a Fourier transform infrared (FTIR) spectrometer (Thermo Fisher Scientific, Inc., Nicolet5700, Madison, WI, USA) for detection.

### 2.8. Scanning Electron Microscopy (SEM)

Scanning electron microscopy (SEM) (American FEI Company, Tecnai G2F 20, Hillsboro, OR, USA) was used to obtain micrographs of the samples before and after printing, at an accelerated voltage of 20 kV. Before the measurement, the printed material was freeze-dried, fixed on the sample holder, and sprayed with gold. The material was enlarged from different directions of the cross and longitudinal sections, and SEM images with a magnification of 500 times were selected to observe the microstructure of the samples. The most representative micrographs were selected for sample analysis [21].

### 2.9. Swelling Tests

The lyophilized samples (0.1 g) were accurately weighed (the mass was denoted as M_0_) and then placed in a 50 mL centrifuge (Shanghai Anting Scientific Instrument Factory, TDL-40B, Shanghai, China) tube. The masses of the samples and centrifuge tubes were denoted as M_1_. Subsequently, 30 mL of distilled water was accurately added with a pipette, and the samples were shaken in a water bath at 30 °C for 1 h and placed at 25 ± 2 °C for 1 h. The supernatant was removed by high-speed centrifugation at 4000 rpm for 30 min. The water on the gel surface was dried with a filter paper to a constant weight. The sediment and centrifuge tube at this time were weighed and recorded as M_2_. The parallel experiments were repeated three times. The swelling (%) was quantified using Equation (2) [22].
(5)$$\mathrm{Swelling}=\frac{\left(M_{2}\;–{\;M}_{1}\right)}{M_{0}}.$$

### 2.10. Determination of Storage Properties

After printing, the 3DFP products were divided into several groups and stored in beakers sealed with plastic films at 4 °C and 25 °C in dry environments. The storage period was terminated when the surface of the 3DFP products deteriorated. During the storage period, the polyphenol content was measured every 2 d, and each treatment was repeated three times [23].

### 2.11. Post-Processing of 3DFP Products

We created the best system for the 3D printing and cured it. The post-processing method was slightly modified according to the one described by [24]. Three post-processing methods are used: boiling, steaming, and roasting. The specific methods are as follows.

Boiling: add water in an induction cooker according to a 1:8 ratio of material to water, and set the temperature to 100 °C for 15 min.

Steaming: steam the sheet on the shelf of an induction cooker at 100 °C for 15 min.

Roasting: place the 3DFP products in a microwave oven at 160 °C for 12 min. Flip sides once in a while.

### 2.12. Texture Properties

After post-processing, the product was cut into pieces using scissors to simulate chewing with teeth. The hardness and chewability of the 3DFP products after post-processing were measured using a texture meter (TA-XT 2, Stable Micro Systems, Ltd., Surrey, UK); the test method is the same as in method Section 2.4. The remaining 3DFP products were stored for future use.

### 2.13. Total Color Change Rate

The printed AM/MC gel was cut into round cakes with a radius of 1.50 cm, and the luminance (L*), red and green (a*), and yellow and blue (b*) of the 3DFP products were measured and recorded using a WSD-3C whiteness meter (Background Kangguang Instrument Co., LTD, WSD-3C, Beijing, China). After post-processing of the gel, the L_0_*, a_0_*, and b_0_* values of the gel were detected again and recorded. The total color change rate ΔE was calculated according to the following formula [25]:(6)$$\Delta E=\sqrt{{\left(L^{*}-L_{0}{}^{*}\right)}^{2}+{\left(a^{*}-a_{0}{}^{*}\right)}^{2}+{\left(b^{*}-b_{0}{}^{*}\right)}^{2}},$$

### 2.14. In Vitro Digestion Simulation

The in vitro digestion simulation was based on an international consensus [26], with slight modifications.

#### 2.14.1. Preparation of Simulated Digestive Fluid Reserve

Simulated saliva, gastric juice, and intestinal juice were prepared according to the method of Minekus [26].

#### 2.14.2. Simulated Oral Digestion In Vitro

A 0.50 g sample of the cured 3DFP products was placed in 50 mL centrifuge tubes. In addition, 3.50 mL simulated saliva, 0.50 mL amylase (enzyme activity of 1500 U/mL), 25 μL 0.30 M CaCl_2_, and 975 μL distilled water were added to the centrifuge tubes and mixed thoroughly. The cured 3DFP products were digested in a 37 °C constant temperature oscillator (Changzhou Wanfeng Instrument Manufacturing Co., Ltd., WHY-2, Changzhou, China) for 2 min. After digestion, the reaction was terminated with liquid nitrogen and stored for later use.

#### 2.14.3. Simulated Gastric Digestion In Vitro

The residual 3DFP products after simulated oral digestion were added to 50 mL centrifuge tubes in the gastric digestion group, where 7.50 mL of simulated gastric juice, 1.60 mL of 25,000 U/mL pig pepsin, 5 μL of 0.30 M CaCl_2_, 702 μL of 5 M HCl, and 193 μL of distilled water were added into the centrifuge tubes. Simulated gastric juice was used as a blank control. The 3DFP products were digested in a constant temperature oscillator at 37 °C for 2 h and sampled at 0, 24, 48, 72, 96, and 120 min. The reaction was terminated immediately after the end of the reaction in liquid nitrogen.

#### 2.14.4. In Vitro Simulated Intestinal Digestion

The residual 3DFP products, after the gastric digestion simulation, were added to 50 mL centrifuge tubes of the intestinal digestion and blank control groups. Intestinal digestion group: 11 mL simulated intestinal fluid, 5 mL 800 U/mL trypsin, 2.50 mL 4% pig bile, 40 μL 0.30 M CaCl_2_, 460 μL 5 M NaOH, and 1 mL distilled water. Simulated intestinal fluid was used as a blank control. The 3DFP products were digested in a constant temperature oscillator at 37 °C for 2 h and sampled at 0, 24, 48, 72, 96, and 120 min. Finally, the centrifuge tubes were quickly placed in liquid nitrogen to terminate the reaction and were subsequently set aside.

The release solution was centrifuged at 3000 rpm for 10 min, the supernatant was collected, and the absorbance of the release solution at 765 nm was measured. According to the change in polyphenol concentration in the release system, the cumulative release rate (Q, %) of polyphenols from the 3DFP products at different times (t, min) was calculated. It is calculated as follows [27]:(7)$$Q\;(\%)=\frac{M_{t}}{M}\;\times \;100\%,$$
where M_t_ is the cumulative AMP mass released at time t and M is the total AMP mass initially embedded in the gel.

### 2.15. Statistical Analysis

SPSS 17.0 and Origin 2018 software were used to analyze the test results. The recorded data are presented as the mean ± standard error (SE) for each analysis. To investigate the significant differences between the data points, analysis of variance (ANOVA) was used (*p* ≤ 0.05).

## 3. Results and Analysis

### 3.1. Effects of MC, PA, and HA Concentrations on the 3DFP Formability and Deviation

In the experiment, varying concentrations of the components (MC, PA, and HA) in the AM/MC gel were tested to determine the optimal formulation for 3D food printing (3DFP). As shown in Figure 1 and Table 1, a 13% MC concentration resulted in poor molding performance and low printing accuracy. When 14% MC was used without adding PA or HA, the mixture became too thick for printing.

However, when 14% MC was combined with 1% PA and 1% HA, the mixture could be printed and formed effectively. The best printing outcome was achieved with this combination (Figure 1, Figure 2, Figure 3, Figure 4, Figure 5, Figure 6, Figure 7 and Figure 8), with a width deviation of only 4.19%, making it more accurate than the yam starch gel used in a study by Wu et al. [28].

The success of the formulation can be attributed to the interaction between MC, PA, and HA. While MC plays a significant role in printability, the wetting effect of HA [29] and the synergistic interaction between PA and HA [30] improve printing accuracy and reduce deviation.

Increasing the concentration of MC, PA, or HA beyond the successful formulation led to poor molding performance due to excessive thickness or poor product support. The results indicate that the optimal formulation for 3DFP using the AM fruit pulp-based hydrogel consists of 14% MC, 1% PA, and 1% HA, displaying good balance in printability, formability, and accuracy.

### 3.2. Effects of MC, PA, and HA Concentrations on Texture Characteristics of Samples

In texture analysis, several factors are used to characterize the properties of 3D food printed samples: hardness—the force needed to cause a shape change to the sample; elasticity—the degree to which the sample can recover its shape between the first and second compressions; adhesion—the force required to separate the sample surface from other attached objects (e.g., tongue, teeth, or mouth); and degree of mastication (chewiness)—the energy required to chew a solid sample to a state sufficient for swallowing [31].

As shown in Table 2 and Table 3, the different formulations impact these properties: Increasing MC content led to higher hardness, elasticity, and chewiness, while adhesion decreased. Adding PA improved hardness, elasticity, and chewiness, and reduced adhesion due to its good gel properties, enhancing the appearance and support ability of the 3DFP products. Adding HA slightly decreased hardness, elasticity, and chewiness, but increased adhesion due to its lubricating and moisturizing qualities.

The optimal formulation—consisting of 14% MC, 1% PA, and 1% HA—demonstrated the highest hardness and elasticity, the lowest adhesion, and the best molding during printing. This combination allowed for successful 3DFP, while providing a balance between texture properties and nutritional value. Overall, the texture analysis reveals how the different components (MC, PA, and HA) influence the performance of the resulting 3DFP products, highlighting the importance of selecting the proper formulation for achieving the desired printing outcomes and product quality.

### 3.3. PCA

The physical properties of each AM printing product, including hardness, elasticity, chewability, and adhesion, were taken as the analysis object, and the principal component analysis of nine kinds of different AM printing products was carried out by using SPSS software [17].

The software analysis results are shown in Table 4. It can be seen from Table 4 that there is one principal component whose eigenvalue is greater than 1. When extracting the first principal component, its variance contribution rate is 79.769%, close to 80%, indicating that this principal component can basically reflect the information of all indicators and can replace the four selected indicators, successfully achieving the purpose of dimension reduction.

The factor loading in Table 5 reflects the contribution rate of each index to the principal component. From the table, it can be observed that hardness, elasticity, chewability, and adhesiveness had higher loadings on the first principal component. This indicates that the first principal component mainly reflects the information of these four indices, with adhesiveness having the largest loading.

Combining Table 4 and Table 5, the weights of the four indices for evaluating the performance of AM printing products were determined, and are presented in Table 6. The weight of hardness is 0.2552, the weight of elasticity is 0.2156, the weight of chewability is 0.2524, and the weight of adhesion is 0.2768. Using Formula (4), the sum score of AM printing products can be obtained from Formula (8):S = 0.2552P_1_ + 0.2156P_2_ + 0.2524P_3_ + 0.2768P_4_(8)

By combining Equations (2) and (3), membership values of the 9 different AM printing products were calculated (Table 7). The comprehensive score of physical properties of AM printing products can be obtained by using Formula (4), which is shown in Table 8. Among them, 8-A (AM gel system with 14% MC, 1% PA, and 1% HA added) achieved the highest score of 0.997. This indicates that the 8-A AM printing product exhibited the best overall performance in terms of its physical properties.

To further investigate the printability and applicability of the different formulations, six printing systems with high scores and varying levels of MC, PA, and HA were selected for study. These systems included four groups with better printability and texture properties and two groups featuring different combinations of PA and HA. The mass ratios of AM pulp, MC, PA, and HA in these groups were as follows: 100:13:0:0 (Table 1, No. 2, 0.670 points); 100:17:1:0 (Table 1, No. 4, 0.842 points); 100:17:0:1 (Table 1, No. 6, 0.532 points); 100:14:1:1 (Table 1, No. 8, 0.997); 100:14:1:2 (Table 1, No. 9, 0.655); and 100:14:2:1 (Table 1, No. 10, too thick for printing).

Each group was evaluated before and after 3D food printing to assess its printability and applicability. By comparing the performance of these printing systems, the study aimed to identify the best formulation for achieving optimal printing results, texture properties, and overall quality in the final 3DFP products.

### 3.4. Effects of MC, PA, and HA Concentrations on Rheological Characteristics of Samples

The dynamic viscoelasticity of samples is crucial for their practical application properties in 3D food printing. There are three main parameters in dynamic viscoelasticity: energy storage modulus (G′)—represents the elasticity and rigidity of the printed material. Higher G′ values indicate better support and reduced likelihood of collapse [32]; loss modulus (G″)—reflects the viscosity of the print [33]; and tanδ—represents fluidity and helps to differentiate between elastic and viscous behaviors.

Both G′ and G″ showed some dependence on frequency, indicating a relationship between frequency and the viscoelastic properties of the gel system. As the frequency increased, the G′ and G″ values for the AM gel systems also showed varying degrees of increase. In all samples, G′ was higher than G″, which demonstrates that the prints were primarily elastic within the scanned frequency range and had a reticular gel structure with a certain level of strength.

Understanding the dynamic viscoelasticity of the samples is important for optimizing the formulation and printing parameters in 3D food printing. By considering the energy storage modulus, loss modulus, and fluidity, researchers can better control and predict the printed products’ performance and texture properties.

Figure 2 shows the differences in the energy storage modulus (G′) and loss modulus (G″) for the 3DFP products before and after printing. Some key observations from this data include the following: G′ values increased and G″ values decreased after 3D printing extrusion, indicating an increase in hardness and elasticity, and a decrease in adhesion for the gel system. The addition of PA increased G′ (elasticity) and decreased G″ (adhesion), due to its good gel characteristics and its interactions with AM pulp and MC that reduce gaps in the system. The addition of HA led to decreased elasticity and increased adhesion because it functions as a lubricant. When PA and HA were added simultaneously, the AM gel system exhibited maximum G′, while the difference between G′ and G″ also reached its peak. This resulted in a stable gel structure with good formability, making the material more suitable for printing.

Increasing the amount of PA led to higher elasticity, lower adhesion, and improved gel strength but reached a point where it could not be printed. On the other hand, increasing HA resulted in lowered elasticity and increased adhesion, negatively impacting the printing effect.

The trend of tanδ decreasing with frequency suggests a reduction in the mobility and viscoelasticity of the 3DFP products, which leads to an increase in their solid-like properties.

In conclusion, to achieve the desired product type after processing, different composition ratios should be carefully selected based on the required solid and liquid states of the 3D printing products. By considering the dynamic viscoelastic properties and subsequent processing requirements, 3D printed products can be optimized for both performance and quality.

### 3.5. Effects of MC, PA, and HA Concentrations on FTIR of Samples

Fourier Transform Infrared Spectroscopy (FTIR) is an effective method for studying the structures of copolymers, as it can measure the vibration of functional groups and polar bonds in samples, allowing the analysis of changes in their chemical bonds [20]. In this study, FTIR was used to investigate the effects of 3DFP on the ink components and the interactions between them before and after adding PA and HA, including the potential formation of covalent bonds.

As shown in Figure 3, both 3DFP materials and 3DFP products exhibit a strong, wide absorption band at 3200–3500 cm^−1^ (peak a), generated by the stretching vibration of phenolic hydroxyl O-H in AM, and a strong absorption band at 1200–1000 cm^−1^ (peak h) produced by the tensile vibration of phenolic C-O in AM. The addition of PA and HA had minimal effects on these peaks, indicating that they did not significantly impact AMP.

Note: (a–m) denote 3386.87, 2931.58, 2063.57, 1724.28, 1645.03, 1449.52, 1409, 1075.95, 947.47, 890.45, 818.49, 776.52, 615.10 cm^−1^, respectively.

Upon adding PA, the peak at 1724.28 cm^−1^ (peak d) became more pronounced, suggesting an enhancement of the C=O stretch of ketone in the 3DFP products. However, no new peaks formed, meaning PA and AM binding was weak. The addition of HA led to the formation of a new peak at 2063.57 cm^−1^ (peak c), implying a new covalent bond was formed between HA and the AM/MC gel.

When PA and HA were added simultaneously, a new absorption peak emerged at 1409 cm^−1^ (peak g), indicating a covalent bond formed between PA and HA. This bond was stronger with the addition of 1% PA and 1% HA. Comparing before and after printing, the bond strength of the 3DFP products increased, possibly due to 3DFP making the sample structure more compact.

### 3.6. Effects of MC, PA, and HA Concentrations on the SEM Images of the Samples

The microstructural analysis of the 3DFP materials and products is illustrated in Figure 4. When PA or HA is added to the AM gel containing MC, the pores become closer due to the gelling and thickening effects of PA and HA, respectively. However, HA’s wetting effect leads to unevenly distributed pores and a relatively loose structure, resulting in lower printing accuracy.

In contrast, the system containing both PA and HA has more uniform and smaller pores. The 3DFP products with 1% PA and 1% HA exhibit the most compact and well-proportioned pore structure, allowing the printed gel to maintain its shape without collapsing. This stability might be attributed to the covalent bonds formed between PA and HA.

When more PA is added to the system, there are fewer pores, but it becomes unprintable due to the high gel performance, which interrupts the material during the printing process. On the other hand, increasing the HA content leads to larger porosity and reduced uniformity in the 3DFP products, possibly due to its high wettability causing uneven discharge, severe deformation, and poor support. These observations are consistent with the results reported by Wang et al. [16].

Additionally, after 3DFP, the pores in all materials significantly reduced, and the structure became more compact, which aligns with previous findings in the study.

### 3.7. Effects of MC, PA, and HA Concentrations on the Swelling Degree of the Samples

Swelling refers to the phenomenon where a colloidal gel dissolves in a solution, and the volume increases as the colloid absorbs the solution. It is an important property of gels, and the degree of swelling can be used as a measure of this property. The swelling characteristics of different gels are related to the degree of crosslinking; the higher the degree of crosslinking, the lower the swelling degree of the gel under the same conditions [34].

As shown in Figure 5, after adding PA, the swelling degree of the printing material decreased because it has a gelling effect that makes the sample structure more compact. When HA was added, the swelling degree of the samples increased due to HA’s wettability. Compared to other systems, the swelling degree of the printing material decreased after simultaneously adding 1% PA and 1% HA. This may be attributed to the formation of covalent bonds between PA and HA, resulting in increased gel crosslinking.

Additionally, the water retention of all materials after printing was better than before printing, likely because the 3DFP process made the sample structure more compact.

In conclusion, the gel system containing 14% MC, 1% PA, and 1% HA had the highest score (0.997), the most stable structure, the best printing effect, the best indicators, and the ideal printability and applicability. Therefore, this proportion of printing material was selected for the storage, post-processing, and intestinal digestion simulations.

### 3.8. Effect of Storage Time on AMP Content of 3DFP Products in Optimal System

The quality of the best 3DFP products during storage was investigated. As shown in Figure 6, the 3DFP product of the optimal system could be stored at 4 °C and 25 °C for 14 d and 6 d, respectively. Because of the high nutrient content of the printing material, the storage time was relatively short. During the storage period, the AMP content of the prints supplemented with 14% MC, 1% PA, and 1% HA gradually decreased at 4 °C and 25 °C. At the end of storage, AMP content decreased by 39.90% and 46.63%, respectively.

### 3.9. Effect of Different Post-Processing Methods on Chromatic Aberration and Texture Properties of 3DFP Products of Optimal System

Figure 7 and Figure 8 show the 3DFP product of the optimal system after post-processing and its chromatic aberration before post-processing, respectively. The results indicate that steaming had a negligible effect on color. However, under boiling treatment, the color was lighter than before post-processing due to the solubility of anthocyanins and other color substances in water.

Compared with the 3DFP products before post-processing, the color difference and color intensity of the 3DFP products after roasting were greater, possibly due to the Maillard reaction, which imparts a pleasing flavor and color. Investigating the biological activities and other effects of roasting requires further research.

Table 9 lists the textural properties of 3DFP in the best post-processing system. Baked 3DFP products had high chewiness and hardness, potentially because the water loss in the baked system was the highest. Cooking chewiness and hardness were lower, likely due to the water-based treatment resulting in less water loss for the 3DFP product.

These findings demonstrate that different post-processing methods have varied effects on the color, hardness, and chewiness of the 3DFP products. The appropriate post-processing method can be selected based on specific requirements and desired product characteristics.

### 3.10. In Vitro Gastrointestinal Digestion Simulation of 3DFP Products of Optimal System

Figure 9 shows that during the 120 min gastric digestion and 120 min intestinal digestion, the cumulative release rate of AMP content under each treatment gradually increased. This indicates that the 3DFP products have sustained release characteristics in simulated gastric juice and simulated intestinal juice.

During gastric digestion, the cumulative release rates of boiling, steaming, and roasting were 8.04%, 15.12%, and 24.79%, respectively, while during intestinal digestion, they were 73.87%, 71.64%, and 72.25%, respectively. This suggests that AMP release characteristics were better in the intestinal simulation solution after sample post-processing, similar to the findings of Bermúdez-Soto et al. [35].

After 120 min of gastric digestion and 120 min of intestinal digestion, the sum of the cumulative release rates under boiling, steaming, and roasting treatments reached 81.91%, 86.76%, and 97.04%, respectively. The roasting treatment had a significantly higher sum of cumulative release rates than other groups, potentially due to the lower temperature causing less damage and less influence on AMP content. Compared to the roasting treatment, the boiling treatment temperature was the same, but part of the water-soluble AMP was lost due to being dissolved in water.

The results show that the gastric environment has a promotional effect on the release of AMP, possibly because AMP contains abundant weakly acidic phenolic hydroxyl groups [36] and is relatively stable under acidic conditions. The acidic environment of gastric juice may cause polyphenol hydrolysis of the bound state in AM, resulting in AMP dissociation [37]. Additionally, some polyphenols undergo hydrolysis under acidic conditions, and part of the glycoside compounds are converted into aglycones, increasing the release rate of AMP [38]. These results are consistent with [39].

AMP is better retained in the intestine, and its release in intestinal digestion simulation is greater than its degradation. This could be due to the hydrolyzing action of pancreatic juice and bile on the chemical bonds that react with AMP, and the stability of some neutral polyphenols in the intestinal simulation solution at pH 7, allowing AMP to be better retained. These findings align with those reported by Rao et al. [39].

In vitro release of AMP content is commonly fitted using several kinetic models, such as the zero-order kinetic model, first-order kinetic model, Niebergall square root equation, and Ritger–Peppas equation model [40,41]. In this study, Origin software was employed to fit four models of samples after 3D printing and curing post-processing to establish the kinetic release equation of AMP content in simulated gastric juice and simulated intestinal juice; the results are presented in Table 10.

According to the correlation coefficient R^2^ results, the in vitro release of AMP from 3DFP products of the best system closely adhered to the Ritger–Peppas equation model fitting. The coefficient n of lnQ = nlnt + k was greater than 0.89, with the exception of the intestinal digestion simulation of the roasting treatment, indicating that the release mechanism was mainly skeleton dissolution. However, the coefficient of intestinal digestion in the roasting treatment fell between 0.45 and 0.89, suggesting that the release mechanism was primarily a combination of diffusion and skeleton dissolution [42].

## 4. Conclusions

In this study, a gel system capable of loading AMP and suitable for 3D printing was developed using AM pulp as the main material, MC as the auxiliary material, and PA and HA as functional additives. The optimal ratio was determined to be AM pulp:MC:PA:HA = 100:14:1:1. This system demonstrated high printing accuracy, excellent physical and rheological properties, and achieved ideal 3D printing suitability compared to other proportions and pre-printed samples.

The system featured a compact structure, uniform pores, good support, high degree of cross-linking, and strong water retention. It could also be stored at 4 °C for 14 days, highlighting its potential applicability for 3D printing. After post-processing, the 3DFP products of this system exhibited a favorable AMP release rate and sustained release effect during gastrointestinal digestion, which is consistent with the Ritger–Peppas equation model.

These findings demonstrated that 3D printing positively influenced the printability and applicability of the material, providing a theoretical foundation for the application of 3D printing using fruit pulp as a raw material.

In future research, relevant technical means can be used to modify the materials and verify and broaden the application scope of 3D printing technology in the field of food to meet the market demand for personalized food. Additionally, the use of methylcellulose as a gel-forming thickener to embed other functional substances requires further research. Moreover, methylcellulose can be added to other fruit pulps to form fruit pulp gel, which can be applied to 3D printing. Their printability and applicability in future food value chain applications require further research.

## Figures and Tables

**Figure 1 foods-12-02068-f001:**
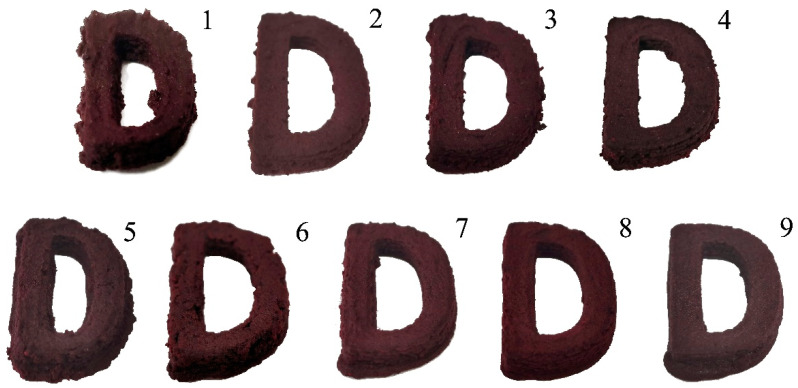
Images of 3DFP products with different mass ratios. Note: Numbers 1–9 in the chart indicate that AM pulp was added according to the mass ratio of MC, PA, and HA in Table 1.

**Figure 2 foods-12-02068-f002:**
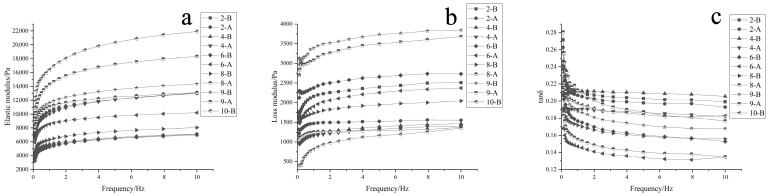
Rheological characteristics of 3DFP materials and products. Note: Numbers 2, 4, 6, 8, 9, and 10 in the chart indicate that AM pulp was added according to the mass ratio of MC, PA, and HA in Table 1. “B” and “A” denote before and after 3DFP, respectively; (**a**–**c**) denote the elastic modulus, loss modulus, and tanδ, respectively.

**Figure 3 foods-12-02068-f003:**
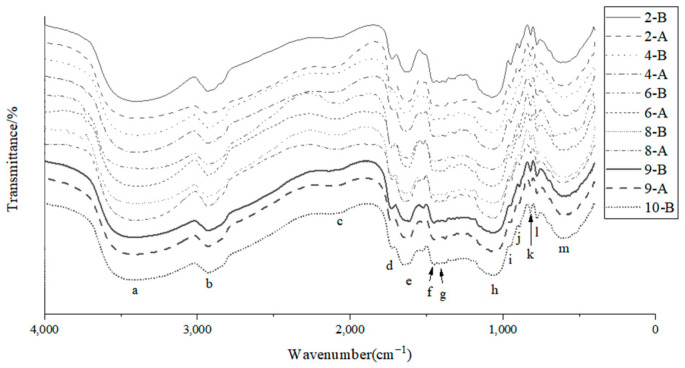
FTIR micrograph of 3DFP materials and 3DFP products.

**Figure 4 foods-12-02068-f004:**
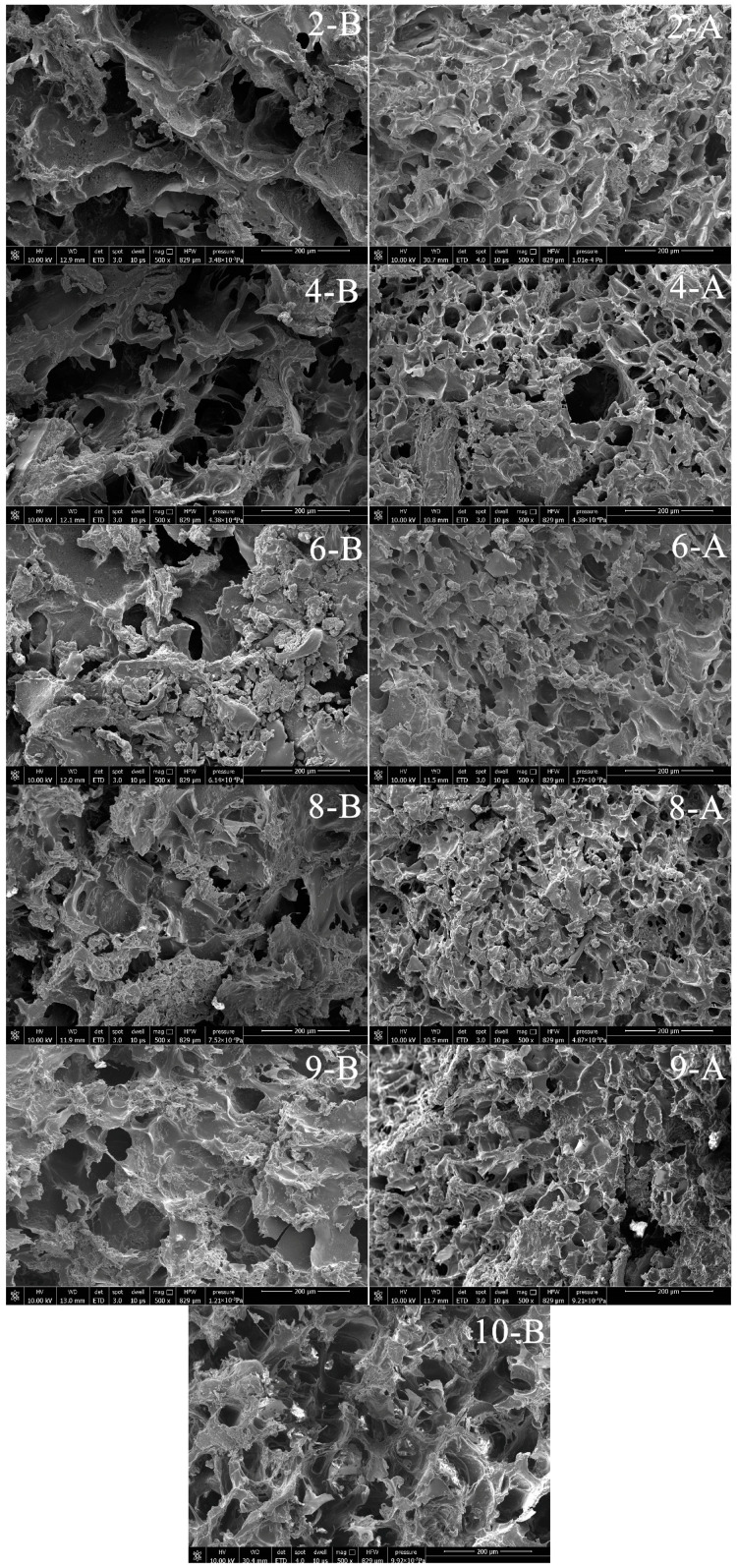
SEM micrograph of 3DFP materials and products.

**Figure 5 foods-12-02068-f005:**
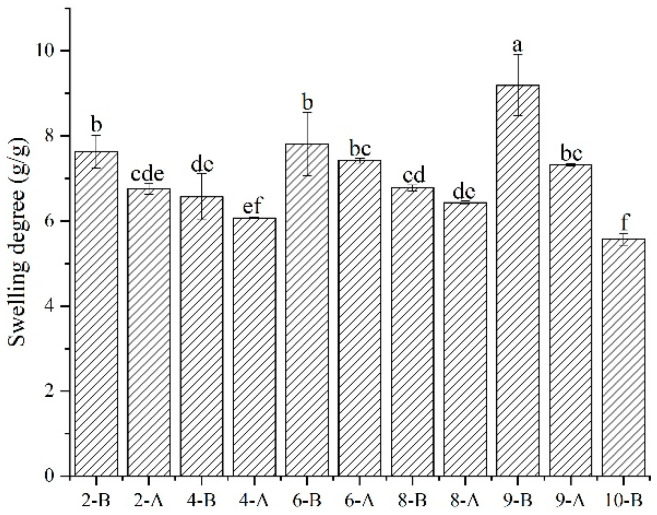
Swelling degree of 3D printing material and 3D printed products. Note: (a–f) Different letters indicate significant difference (*p* < 0.05).

**Figure 6 foods-12-02068-f006:**
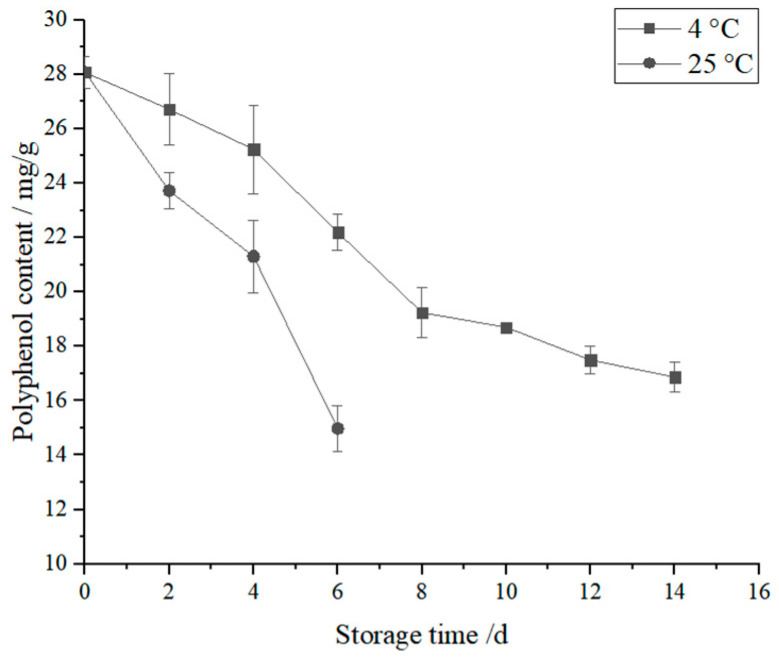
Storage time on AMP content of 3DFP products in optimal system.

**Figure 7 foods-12-02068-f007:**
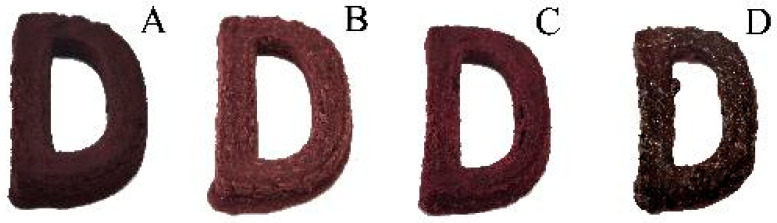
Effects of different post-processing methods on the appearance of 3D products. Note: (**A**–**D**) represent uncured 3DFP products, and 3DFP products after boiling, steaming, and roasting, respectively.

**Figure 8 foods-12-02068-f008:**
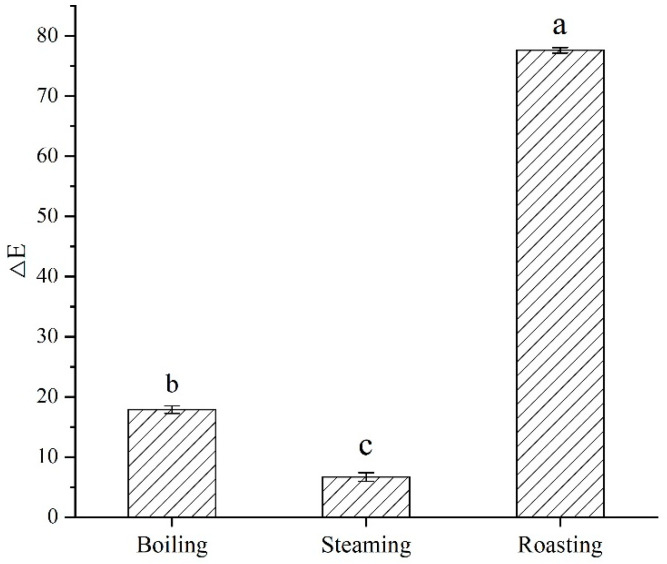
Chromatic aberration of 3DFP products by different post-processing methods. Note: (a–c) Different letters indicate significant difference (*p* < 0.05).

**Figure 9 foods-12-02068-f009:**
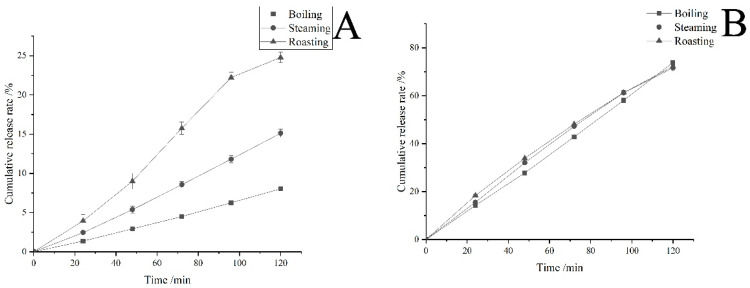
AMP release rate of the best 3DFP products. Note: (**A**) denotes gastric digestion and (**B**) denotes intestinal digestion.

**Table 1 foods-12-02068-t001:** Formability of 3DFP material with different mass ratios.

Formulation	MC/PA/HA	Print Span (mm)	Print Deviation (%)
1	12% MC	7.00 ± 0.32	39.97 ± 6.43
2	13% MC	6.26 ± 0.55	25.21 ± 11.05
3	16% MC/1% PA	5.28 ± 0.20	5.67 ± 4.09
4	17% MC/1% PA	5.24 ± 0.18	4.86 ± 3.50
5	16% MC/1% HA	6.12 ± 0.17	22.48 ± 3.42
6	17% MC/1% HA	5.52 ± 0.38	10.41 ± 7.50
7	13% MC/1% PA/1% HA	5.50 ± 0.22	10.01 ± 4.32
8	14% MC/1% PA/1% HA	5.21 ± 0.23	4.19 ± 4.60
9	14% MC/1% PA/2% HA	6.78 ± 0.34	35.60 ± 6.80
10	14% MC/2% PA/1% HA	/	/
11	14% MC	/	/
12	14% MC/1% PA	/	/
13	14% MC/1% HA	/	/
14	15% MC/1% PA/1% HA	/	/

Note: AM fruit pulp was 100%; MC, PA, and HA were added according to the percentages in the table. The print width of the model design value was 5.00 mm.

**Table 2 foods-12-02068-t002:** Texture characteristics of 3DFP materials with different mass ratios.

Formulation Codes	Hardness (g)	Springiness	Chewiness (g·mm)	Adhesiveness (g·s)
1-B	526.59 ± 42.11 ^e^	0.1821 ± 0.0291 ^bc^	15.09 ± 1.35 ^de^	–135.78 ± 8.01 ^de^
2-B	755.23 ± 62.44 ^c^	0.1893 ± 0.0410 ^b^	24.87 ± 1.63 ^ab^	–103.4 ± 11.16 ^bd^
3-B	646.27 ± 19.55 ^e^	0.1535 ± 0.0369 ^bc^	16.03 ± 2.67 ^d^	–114.78 ± 20.65 ^cd^
4-B	895.45 ± 45.49 ^b^	0.1442 ± 0.0164 ^bc^	20.47 ± 1.92 ^c^	–81.99 ± 14.19 ^b^
5-B	519.58 ± 63.22 ^e^	0.1245 ± 0.0281 ^c^	12.07 ± 2.33 ^e^	–150.72 ± 9.23 ^e^
6-B	711.39 ± 45.38 ^cd^	0.1660 ± 0.0361 ^bc^	27.01 ± 2.29 ^a^	–120.04 ± 16.89 ^cd^
7-B	663.42 ± 36.54 ^d^	0.1941 ± 0.0329 ^b^	22.95 ± 1.23 ^bc^	–107.69 ± 6.79 ^c^
8-B	926.14 ± 34.99 ^b^	0.2753 ± 0.0180 ^a^	23.45 ± 0.99 ^bc^	–83.85 ± 10.33 ^b^
9-B	807.90 ± 71.32 ^bc^	0.1473 ± 0.0193 ^bc^	25.36 ± 2.59 ^ab^	–80.30 ± 13.00 ^b^
10-B	1065.72 ± 58.94 ^a^	0.2533 ± 0.0432 ^a^	32.28 ± 2.66 ^a^	–23.06 ± 14.67 ^a^

Note: Numbers 1–10 in the table indicate that AM pulp was added according to the mass ratio of MC, PA, and HA in Table 1. “B” indicates before 3DFP. Different letters in the same column indicate significant difference (*p* < 0.05).

**Table 3 foods-12-02068-t003:** Texture characteristics of 3DFP products with different mass ratios.

Formulation Codes	Hardness (g)	Springiness	Chewiness (g·mm)	Adhesiveness (g·s)
1-A	686.47 ± 69.54 ^d^	0.1968 ± 0.0350 ^cd^	24.66 ± 2.46 ^b^	–100.80 ± 15.39 ^ab^
2-A	890.92 ± 95.66 ^ab^	0.2055 ± 0.0109 ^bc^	30.68 ± 3.44 ^a^	–81.30 ± 20.47 ^a^
3-A	728.55 ± 58.47 ^cd^	0.2340 ± 0.0412 ^abc^	23.90 ± 0.99 ^b^	–99.22 ± 32.43 ^ab^
4-A	976.32 ± 73.55 ^ab^	0.2430 ± 0.0262 ^abc^	31.65 ± 2.86 ^a^	–71.11 ± 12.77 ^a^
5-A	654.15 ± 77.23 ^d^	0.1228 ± 0.0333 ^e^	18.95 ± 1.93 ^c^	–133.71 ± 9.64 ^b^
6-A	865.59 ± 78.01 ^bc^	0.1496 ± 0.0291 ^de^	33.57 ± 2.34 ^a^	–96.52 ± 23.67 ^ab^
7-A	744.64 ± 92.13 ^cd^	0.2574 ± 0.0294 ^ab^	29.97 ± 2.01 ^a^	–87.96 ± 32.16 ^a^
8-A	1029.98 ± 86.49 ^a^	0.2867 ± 0.0160 ^a^	33.34 ± 1.67 ^a^	–65.92 ± 17.58 ^a^
9-A	913.08 ± 68.40 ^a^	0.2010 ± 0.0538 ^bc^	31.34 ± 3.23 ^a^	–88.16 ± 17.47 ^a^

Note: Numbers 1–9 in the table refer to the mass ratios of AM pulp, MC, PA, and HA in Table 1. “A” indicates after 3DFP. Different letters in the same column indicate significant difference (*p* < 0.05).

**Table 4 foods-12-02068-t004:** Results of contribution rate by principal component analysis.

Components	Characteristic Root	Variance Contribution Rate (%)	Cumulative Variance Contribution Rate (%)
1	3.1910	79.7690	79.7690
2	0.6320	15.8050	95.5740
3	0.1490	3.7330	99.3070
4	0.0280	0.6930	100

**Table 5 foods-12-02068-t005:** Component matrix of principal component analysis.

Components	Hardness	Springiness	Chewiness	Adhesiveness
1	0.908	0.767	0.898	0.985

**Table 6 foods-12-02068-t006:** Normalized data.

Indexes	Normalized Data
Hardness	0.2552
Springiness	0.2156
Chewiness	0.2524
Adhesiveness	0.2768

**Table 7 foods-12-02068-t007:** Membership degree value.

	P_1_ (Hardness Positive Effect)	P_2_ (Springiness Positive Effect)	P_3_ (Chewiness Positive Effect)	P_4_ (Adhesiveness Negative Effect)
1-A	0.0860	0.4515	0.3906	0.4855
2-A	0.6300	0.5046	0.8023	0.7731
3-A	0.1980	0.6785	0.3386	0.5088
4-A	0.8572	0.7334	0.8687	0.9234
5-A	0.0000	0.0000	0.0000	0.0000
6-A	0.5626	0.1635	1.0000	0.5486
7-A	0.2408	0.8212	0.7538	0.6749
8-A	1.0000	1.0000	0.9843	1.0000
9-A	0.6890	0.4771	0.8475	0.6719

**Table 8 foods-12-02068-t008:** Comprehensive score sheet.

Formulation Codes	1-A	2-A	3-A	4-A	5-A	6-A	7-A	8-A	9-A
Synthesis score	0.370	0.670	0.456	0.842	0.000	0.532	0.641	0.997	0.655

**Table 9 foods-12-02068-t009:** Texture properties of 3D printed products treated by different post-processing methods.

Post-Processing Method	Hardness (g)	Chewiness (g·mm)
Boiling	5359.91 ± 59.13 ^c^	1877.61 ± 544.99 ^c^
Steaming	8521.74 ± 988.99 ^b^	3327.78 ± 925.57 ^b^
Roasting	33,988.44 ± 383.69 ^a^	11,399.75 ± 320.79 ^a^

Note: Different letters in the same column indicate significant difference (*p* < 0.05).

**Table 10 foods-12-02068-t010:** The kinetic model fitting of AMP release.

Post-Processing Method	Types of Digestion	Model	Fitted Equation	R^2^
Boiling	Peptic digestion	Zero-order dynamics	Q = 0.06714t − 0.1773	0.9973
		First-order dynamics	ln(100 − Q) = (−6.9938 × 10^−4^)t + 4.6074	0.9957
		Niebergall square root equation	(1 − Q)^1/2^ = −0.0034t + 10.0101	0.8589
		Ritger–Peppas equation	lnQ = 1.0922lnt − 3.1529	0.9998
	Intestinal digestion	Zero-order dynamics	Q = 0.6141t − 0.7201	0.9991
		First-order dynamics	ln(100 − Q) = −0.0108t + 4.7129	0.9410
		Niebergall square root equation	(1 − Q)^1/2^ = −0.0401t + 10.2275	0.8756
		Ritger–Peppas equation	lnQ = 1.0248lnt − 0.6182	0.9991
Steaming	Peptic digestion	Zero-order dynamics	Q = 0.1272t − 0.4115	0.9969
		First-order dynamics	ln(100 − Q) = −0.0014t + 4.6112	0.9936
		Niebergall square root equation	(1 − Q)^1/2^ = −0.0066t + 10.0252	0.8538
		Ritger–Peppas equation	lnQ = 1.1323lnt − 2.7	0.9999
	Intestinal digestion	Zero-order dynamics	Q = 0.6076t + 1.4759	0.9940
		First-order dynamics	ln(100 − Q) = −0.0106t + 4.6732	0.9844
		Niebergall square root equation	(1 − Q)^1/2^ = −0.0396t + 10.0834	0.9179
		Ritger–Peppas equation	lnQ = 0.9598lnt − 0.2788	0.9959
Roasting	Peptic digestion	Zero-order dynamics	Q = 0.2209t − 0.6349	0.9849
		First-order dynamics	ln(100 − Q) = −0.0025t + 4.6173	0.9828
		Niebergall square root equation	(1 − Q)^1/2^ = −0.0118t + 10.0453	0.8560
		Ritger–Peppas equation	lnQ = 1.1872lnt − 2.3824	0.9901
	Intestinal digestion	Zero-order dynamics	Q = 0.6002t + 2.9779	0.9911
		First-order dynamics	ln(100 − Q) = −0.0106 + 4.6560	0.9872
		Niebergall square root equation	(1 − Q)^1/2^ = −0.0393t + 10.0016	0.9390
		Ritger–Peppas equation	lnQ = 0.8561lnt − 0.2002	0.9991

## Data Availability

Data sharing is not applicable—no new data were generated, or the article describes entirely theoretical research.

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
