# Peer review of "Effect of 3D Food Printing Processing on Polyphenol System of Loaded Aronia melanocarpa and Post-Processing Evaluation of 3D Printing Products"

_foods, 2023, doi:10.3390/foods12102068_

Round 1

Reviewer 1 Report

The manuscript was well-designed and it had good results. 

Its introduction, methods and results sections were suitable.

The developed system had a compact structure, good support, a high degree of cross-linking, and good water retention and it was good for 3-D printing. It can be advance research for its field.

Author Response

Dear reviewer:

Thank you for your letter and the reviewers’ comments on our manuscript entitled " Effect of 3D food printing processing on polyphenol system of loaded Aronia melanocarpa and post- processing evaluation of 3D printing products " (ID: foods-2334944). Those comments are very helpful for revising and improving our paper, as well as the important guiding significance to other studies. Thank you very much for your recognition, your recognition is very valuable to our research!

We accepted the suggestion of reviewer and polished English language and style.

Kind regards,

Nan Xijun

Corresponding author: Ruan Zheng

E-mail address: [email protected]

Reviewer 2 Report

The paper submitted is very interesting about the 3D food printing field, but I think some explanations should be improved. 

- The curing methods should be explained and better justified. 

- There are many equipments/instruments/devices that have not been reported either model, trademark, city, country, etc... for example texturometer and centrifuge.  

- The equation 3 quality must be improved. 

- The authors called curing methods and not post-processing methods.

- Diverse figures should improve their definition quality, for example, figure 8. 

- This formulation fruit pulp:methylcellulose:pea albumin: hyaluronic acid = 100:14:1:1, how was it selected? 

Author Response

Dear reviewers:

Thank you for your letter and the reviewers’ comments on our manuscript entitled " Effect of 3D food printing processing on polyphenol system of loaded Aronia melanocarpa and post- processing evaluation of 3D printing products " (ID: foods-2334944). Those comments are very helpful for revising and improving our paper, as well as the important guiding significance to other studies. We have studied the comments carefully and made corrections which we hope meet with approval. The main corrections are in the manuscript and the responds to the reviewers’ comments are as follows (the replies are highlighted in blue).

Replies to the reviewers’ comments:

  1. English language and style are fine/minor spell check required.

    Response: We accepted the suggestion of reviewer and polished English language and style.

  2. The curing methods should be explained and better justified. Response: We accepted the suggestion of reviewer and made corresponding modifications.
  3. There are many equipments/instruments/devices that have not been reported either model, trademark, city, country, etc... for example texturometer and centrifuge.  Response: We accepted the suggestion of reviewer and made corresponding modifications.
  4. The equation 3 quality must be improved. 
    Response: We accepted the suggestion of reviewer and made corresponding modifications.
  5. The authors called curing methods and not post-processing methods.
    Response: We accepted the suggestion of reviewer and made corresponding modifications.

  6. Diverse figures should improve their definition quality, for example, figure 8.
    Response: We accepted the suggestion of reviewer and made corresponding modifications.

  7. This formulation fruit pulp:methylcellulose:pea albumin: hyaluronic acid = 100:14:1:1, how was it selected?
    Response: We accepted the suggestion of reviewer and made corresponding modifications.

    Once again, thank you very much for your constructive comments and suggestions which would help us both in English and in depth to improve the quality of the paper.

    Kind regards,

    Nan Xijun

    Corresponding author: Ruan Zheng

    E-mail address: [email protected]

Reviewer 3 Report

This manuscript entitled “Effect of 3D food printing processing on polyphenol system of loaded Aronia melanocarpa and post- curing evaluation of 3D printing products.” is an interesting and original study.

The paper is clearly presented and results are very useful. However, there are points for improvement.

1.      Text and tables. The number of digits of the error value depends on the place where the significant digit appears and the number of digits of the corresponding data should be adjusted by taking into account the corresponding error values. In this way, each value in the Tables must be expressed with the significant digits according to the significant digits of each error value (in this case the significant digits of the standard deviation). Please correct it.

2.      The resolution of the figures should be higher.

3.      The term swelling does not reflect the test being performed which would be water holding capacity.

4.      In vitro digestion should be by means of a standardised protocol such as INFOGEST.

Author Response

Dear reviewers:

Thank you for your letter and the reviewers’ comments on our manuscript entitled " Effect of 3D food printing processing on polyphenol system of loaded Aronia melanocarpa and post- processing evaluation of 3D printing products " (ID: foods-2334944). Those comments are very helpful for revising and improving our paper, as well as the important guiding significance to other studies. We have studied the comments carefully and made corrections which we hope meet with approval. The main corrections are in the manuscript and the responds to the reviewers’ comments are as follows (the replies are highlighted in blue).

Replies to the reviewers’ comments:

1. English very difficult to understand/incomprehensible.

Response: We accepted the suggestion of reviewer and polished English language and style.

2. Text and tables. The number of digits of the error value depends on the place where the significant digit appears and the number of digits of the corresponding data should be adjusted by taking into account the corresponding error values. In this way, each value in the Tables must be expressed with the significant digits according to the significant digits of each error value (in this case the significant digits of the standard deviation). Please correct it.

Response: We accepted the suggestion of reviewer and made corresponding modifications.

3. The resolution of the figures should be higher.

Response: We accepted the suggestion of reviewer and made corresponding modifications and supplements.

4. The term swelling does not reflect the test being performed which would be water holding capacity.

Response: We accepted the suggestion of reviewer and made corresponding modifications.

5. In vitro digestion should be by means of a standardised protocol such as INFOGEST.

Response: We accepted the suggestion of reviewer and made corresponding modifications. We refer to: "A standardised static in vitro digestion method suitable for food – an international consensus."

Once again, thank you very much for your constructive comments and suggestions which would help us both in English and in depth to improve the quality of the paper.

Kind regards,

Nan Xijun

Corresponding author: Ruan Zheng

E-mail address: [email protected]

Reviewer 4 Report

The paper Effect of 3D food printing processing on polyphenol system of  loaded Aronia melanocarpa and post- curing evaluation of 3D 3 printing products is interesting and worth o investigation. However, the main draw back of the paper is the lack desrpption of analysed materials and methods of preparation.

Some elements should be explained and described:

-Chapter 2.1 is lack of information. The ingrediencies and methods applied to prepare the inks should be detailly described. Composition of printing inks etc. What kind of food printers were used? The formulation codes should be described.

-Chapter 2.6. The final material of this paper is gel. In this chapter Authors mentioned that the powders were used. How the powders were obtained after printing?

Chapter 2.8. What was the condition of freeze-drying of gels?

-Chapter 2.9. The gels after printing were stored maybe  after printing and drying or roasting?

-Chapter 2.11. What kind of test were used to measure  hardness and chewability of this gels

-Figure 2, 8, 9  The readability of Figures should be improved.

-Authors described the applied data separately, the obtained data should be correlated to find the link between them (with application of correlation analysis of PCA test). It will improve the discussion and conclusions.

Author Response

Dear reviewers:

Thank you for your letter and the reviewers’ comments on our manuscript entitled " Effect of 3D food printing processing on polyphenol system of loaded Aronia melanocarpa and post- processing evaluation of 3D printing products " (ID: foods-2334944). Those comments are very helpful for revising and improving our paper, as well as the important guiding significance to other studies. We have studied the comments carefully and made corrections which we hope meet with approval. The main corrections are in the manuscript and the responds to the reviewers’ comments are as follows (the replies are highlighted in blue).

Replies to the reviewers’ comments:

1. Moderate English changes required.

Response: We accepted the suggestion of reviewer and polished English language and style.

2. Chapter 2.1 is lack of information. The ingrediencies and methods applied to prepare the inks should be detailly described. Composition of printing inks etc. What kind of food printers were used? The formulation codes should be described.

Response: We accepted the suggestion of reviewer and made corresponding modifications and supplements.

3. Chapter 2.6. The final material of this paper is gel. In this chapter Authors mentioned that the powders were used. How the powders were obtained after printing?

Response: We accepted the suggestion of reviewer and made corresponding modifications and supplements. The AM gel powders were obtained after lyophilization (-50 °C, vacuum degree is 22) by a lyophilizer (Beijing Boyikang Experimental Instrument Co., LTD., FD-1A-50, Beijing, China), and stored in -20 °C refrigerator for future use. 

4. Chapter 2.8. What was the condition of freeze-drying of gels?

Response: We accepted the suggestion of reviewer and made corresponding modifications and supplements.

5.Chapter 2.9. The gels after printing were stored maybe  after printing and drying or roasting?

Response: We accepted the suggestion of reviewer and made corresponding modifications and supplements. After printing, the 3DFP products were divided into several groups and stored in beakers sealed with plastic films at 4 °C and 25 °C in dry environments. 

6. Chapter 2.11. What kind of test were used to measure  hardness and chewability of this gels.

Response: We accepted the suggestion of reviewer and made corresponding modifications and supplements. The test method is the same as  method  2.4.

7. Figure 2, 8, 9  The readability of Figures should be improved.

Response: We accepted the suggestion of reviewer and made corresponding modifications and supplements.

8. Authors described the applied data separately, the obtained data should be correlated to find the link between them (with application of correlation analysis of PCA test). It will improve the discussion and conclusions.

Response: We accepted the suggestion of reviewer and made corresponding modifications and supplements. 2.5, 3.3 were added.

Once again, thank you very much for your constructive comments and suggestions which would help us both in English and in depth to improve the quality of the paper.

Kind regards,

Nan Xijun

Corresponding author: Ruan Zheng

E-mail address: [email protected]

Round 2

Reviewer 2 Report

The document has been considerably improved with respect to the original document. The document can be accepted as it stands.

Reviewer 3 Report

The paper has been improved. The new version could be accepted. However, there are some figures with low quality.  Authors must be improved them.

Reviewer 4 Report

The paper was improved but still Figure 2 is not readable.